



# Probabilistic soil moisture dynamics of water– and energy–limited ecosystems

Estefanía Muñoz[1,2], Andrés Ochoa[2], and Germán Poveda[2]

[1]Max Planck Institute for Biogeochemistry, Jena, Germany
[2]Department of Geosciences and Environment, Universidad Nacional de Colombia, Medellín, Colombia

**Correspondence:** E. Muñoz (ehoyos@bgc-jena.mpg.de)

**Abstract.** We present an extension of the stochastic ecohydrological model for soil moisture dynamics at a point of Rodríguez-Iturbe et al. (1999) and Laio et al. (2001). In the original model, evapotranspiration is a function of soil moisture and vegetation parameters, which makes the model suitable for water–limited environments. Based on the Leuning's stomatal conductance approach, the $C_3$ photosynthesis model of Farquhar et al. and the Penman–Monteith equation, we model daily transpiration as a negative exponential function of available photosynthetically active radiation. This function allowed us to broaden the Rodríguez-Iturbe et al. (1999) and Laio et al. (2001) model to encompass both water– and energy–limited ecosystems by introducing the dependence of maximum evapotranspiration on available photosynthetically active radiation. We illustrate the extended model with two study cases from the FLUXNET database, DE–Hai in Germany and GF–Guy in French Guiana, and analyze the sensibility of soil moisture dynamics and the long-term water balance to available radiation. Our results show that the analytical solution presented by Rodríguez-Iturbe et al. (1999) continues to be valid as the maximum evapotranspiration rate is calculated in terms of available energy and assuming stationary in the radiation regime.

## 1 Introduction

Soil water content is a key player in the climate–soil–vegetation system (Entekhabi and Brubaker, 1995; Porporato and Rodríguez-iturbe, 2002; Rodríguez-Iturbe and Porporato, 2004). This system involves a large suite of variables and processes with high spatial and temporal variability, feedbacks and nonlinear relations. Furthermore, soil moisture depends critically on the physiological characteristics of vegetation, pedology and climate (Entekhabi and Rodríguez-Iturbe, 1994; Rodríguez-Iturbe et al., 1999; Rodriguez-Iturbe et al., 2001; Mimeau et al., 2021). Climate and weather patterns determine the amount of water and energy available, crucially impacting the evapotranspiration process (Leuning, 1995; De Pury and Farquhar, 1997; Stoy et al., 2009; Manzoni et al., 2011). Soil texture, its mineralogical composition, and the particle size distribution determine the storage capacity of the soil. Vegetation controls the energy and water fluxes, linking the soil and the atmosphere (Feddes et al., 2001; Rodriguez-Iturbe et al., 2001).

Climate, soil, and vegetation are related through physical, chemical and biological processes, which control the mass and energy transport between land and atmosphere (Eagleson, 1978), while actual evapotranspiration couples water and energy balances. There are two evapotranspiration ($ET$) regimes related to soil moisture: an energy-limited regime and a water-



limited regime. Between these two regimes, there are seasonal environments, in which the availability of water and energy fluctuates.

Among the approaches to modeling soil moisture are biophysical process-based, physical-based and statistical models (Wang et al., 2019). These models mostly feed on in situ (e.g. Korres et al., 2015; Noh et al., 2015; Pirone et al., 2015; Gevaert et al., 2018) and remote sensing (e.g. Wagner et al., 1999; Kim and Barros, 2002; Fang and Lakshmi, 2014; Zehe et al., 2018) data or

involve numerical simulations (e.g. Mtundu and Koch, 1987; Brubaker, 1995; Brubaker and Entekhabi, 1996; Albertson and Montaldo, 2003; Ridolfi et al., 2003; Rigon et al., 2006; Margulis and Entekhabi, 2001; Sela et al., 2012; Chen et al., 2017; de Assunção et al., 2018). In situ data cannnot be extrapolated to regional scales owing to self-organization and nonlinear emergent phenomena arising from such complex processes, and remote sensing methods measure high-resolution spatiotemporal information but only comprise the most superficial layers of the soil (Niemann, 2004), and numerical simulations do not permit

to generalize the results (Ogren, 1993). Daly and Porporato (2005), Seneviratne et al. (2010), Asbjornsen et al. (2011), Legates et al. (2011) and Wang et al. (2019) present some complete reviews of the state of the art of soil moisture modeling.

Eagleson (1978), Cordova and Bras (1981), Hosking and Clarke (1990), and Milly (1993) initiate a biophysical based approach that comprises simplified but realistic conceptual models that analytically describe the phenomena involved in the climate-soil-vegetation system. This approach involves stochastic components that take into account the randomness of precip-

itation and the inherent variability of soil and vegetation properties. Some models have been developed following this approach (e.g. Rodríguez-Iturbe et al., 1999; D'Odorico et al., 2000; Laio et al., 2001; Milly, 2001; Laio et al., 2002; Porporato et al., 2003; D'Odorico and Porporato, 2004; Daly and Porporato, 2006; De Michele et al., 2008; Laio et al., 2009; Rodriguez-Iturbe et al., 2021), modeling precipitation as a stochastic process and deriving analytical expressions of soil moisture dynamics from the soil, climate and vegetation parameters. Such models have been developed for arid and semiarid environments, character-

ized by scarce rainfall, low soil moisture, recurrent water stress, and deep water table (Laio et al., 2009). Since the available energy is not directly considered, they are not suitable to be applied in energy-limited environments.

Photosynthetically active radiation (PAR) is the energy source of biophysical processes, such as photosynthesis, stomatal conductance, transpiration, evaporation, leaf temperature, plant growth, seedling generation, volatile organic compounds emissions, biochemical cycling, and atmospheric chemistry (Thorpe et al., 1978; Baldocchi and Meyers, 1991; Baldocchi and

Collineau, 1994; Ballaré, 1994; Hansen, 1999; Yu et al., 2004; Daly et al., 2004; Ge et al., 2011; Gu et al., 2017), which are directly or indirectly related to soil moisture. On the other hand, the stomata movement regulates simultaneously the water and $CO_2$ fluxes during transpiration and photosynthesis (Collatz et al., 1991; Yu et al., 2004; Medlyn et al., 2017; Shan et al., 2019), being necessary to model photosynthesis and transpiration coupled with the stomatal conductance ($g_s$).

In this study, we introduce an extension of the model by Rodríguez-Iturbe et al. (1999) and Laio et al. (2001) to represent

the stochastic behavior of soil moisture in both water– and energy–limited environments. The moisture loss model proposed by Laio et al. (2001) is modified in such a way that actual $ET$ becomes a function of soil moisture and available radiation. Then, we analyze the relations of transpiration ($T$) and available radiation, and transpiration and soil moisture when radiation is the limiting variable. Stomatal conductance is modeled using the Leuning's approach (Leuning, 1990, 1995), and transpiration using the Penman–Monteith equation. Net assimilation of $CO_2$ ($A_n$) is determined with the Farquhar model and information from





two sites from the FLUXNET database, relating $T$ and PAR through a simple expression. Finally, we analyze the sensitivity of the probability density distribution (pdf) to the available energy and the long-term water balance.

The work is distributed as follows: Section 2 reports the variables taken from the FLUXNET dataset and the values of the parameter used for the stomatal, transpiration and $C_3$ photosynthesis models, Section 3 reviews important remarks on transpiration dynamics in water– and energy–limited environments and describes the models used to relate transpiration and available energy. Section 4 summarizes the soil moisture model of Rodríguez-Iturbe et al. (1999) and Laio et al. (2001) and its extension. Section 5 shows the results aggregated at the daily level and proposes a relationship between transpiration and PAR, while Section 6 analyzes the response of soil water dynamics to high and low values of radiation using four dimensionless groups, and Section 7 focus on the response of the water balance components to low and high values of available energy. Finally, Section 8 outlines the main conclusions.

## 2   Data

Half hourly resolution data of air temperature ($\theta_a$), atmospheric pressure ($P_a$), vapor pressure deficit ($D$), photosynthetic photon flux density (PPFD), net ecosystem $CO_2$ exchange (NEE), $CO_2$ air concentration ($c_a$), and soil moisture ($s$) in two sites in Germany (DE–Hai) and French Guiana (GF–Guy) (Baldocchi et al., 2001; Olson et al., 2004). Table 1 shows the parameters for applying the Penman–Monteith and Leuning equations, and Table 2 those to use the Farquhar model. These values are the same published by Daly et al. (2004).

## 3   Transpiration dynamics

Evaporation and transpiration are key components of the hydrological cycle over land. Their analysis and understanding are fundamental in applications associated with water budgets, biogeochemical cycles, nutrient losses, salt accumulations in soils, production efficiency, etc. (Schulze et al., 1995). Transpiration couples water and carbon cycles (Miner et al., 2017; Shan et al., 2019), while evapotranspiration couples water and land–surface energy balances (Fisher et al., 2009; Seneviratne et al., 2010; Zhang et al., 2016). Such processes are driven by two-way feedbacks between soil, vegetation, atmosphere and climate (Bedoya-Soto et al., 2018). For instance, sensible and latent heat fluxes from vegetation affect the dynamics and thermodynamics of the atmospheric boundary layer and, at the same time, vegetation responds to changes in air temperature and humidity (Monteith and Unsworth, 2013). Vegetation closes its stomata in absence of light or water in the soil so that both radiation and soil moisture are variables directly related to transpiration (Monteith, 1995).

Although transpiration ($T$) responds to a wide variety of complex environmental and physiological factors (Cowan and Farquhar, 1977; Tuzet et al., 2003), here we assume that $T$ can be limited by three factors: soil water, energy, and vegetation capacity (physiology) (see Fig. 1). Soil moisture ($s$) is quantified as the ratio of water volume content to soil porosity. The maximum rate at which vegetation can transpire (with no external limitations) depends on the maximum stomatal conductance, which is directly proportional to pore width (Larcher, 1995). This rate is $T_{maxmax}$ and is represented by the red line in Fig. 1.



**Table 1.** Parameters for the stomatal and transpiration models.

| Parameter | Value | Description |
|---|---|---|
| $a_1$ | 18 | Eq. 3 |
| $c_a$ [$\mu$mol mol$^{-1}$] | 350 | Atmospheric $CO_2$ concentration |
| $c_p$ [J kg$^{-1}$ K$^{-1}$] | 1013 | Specific heat of air |
| $D_x$ [kPa] | 0.3 | Eq. 3 |
| $e$ | 0.622 | Ratio molecular weight of water vapour/dry air |
| $g_a$ [mm s$^{-1}$] | 20 | Atmospheric conductance |
| $g_b$ [mm s$^{-1}$] | 20 | Leaf boundary layer conductance |
| LAI [m m$^{-1}$] | 1.4 | Leaf area index |
| $\lambda_w$ [J kg$^{-1}$] | $2.26 \cdot 10^6$ | Latent heat of water vaporization |
| $\rho_a$ [kg m$^{-3}$] | 1.2 | Air density |
| $\rho_w$ [kg m$^{-3}$] | 997 | Water density |

The left panel of Fig. 1 shows the relationship between transpiration rate and available radiation ($R$) under no water limitations (green line). This relationship is direct until a value of $R$ where transpiration tappers off. This dependence is analyzed in detail in Section 3.3. The right panel in Fig. 1 shows the relationship between transpiration and soil moisture. The dark blue line shows the behavior of transpiration limited by soil moisture and vegetation physiology, but not by energy. Transpiration is maximum for values of $s$ greater than the incipient stomata closure ($s^*$) ($T$ is equal to $T_{maxmax}$). For values lower than $s^*$, $T$ starts decreasing because vegetation closes its stomata to avoid internal losses of water. Transpiration continues to reduce until the wilting point ($s_w$) where it becomes zero. When considering both water and energy limitations, energy limits transpiration for values above $s^*$ (see the plateau of the right panel in Fig. 1), while soil moisture is the limiting factor for values below $s^*$ (Petersen et al., 1992).

High values of $R$ result in higher maximum transpiration rates ($T_{max}$). For example, as shown by the light blue lines in Fig. 1, high values of available energy value ($R_1$) result in a higher transpiration rate for $s > s^*$ ($T_{max1}$) than lower values ($R_2$) associated with lower transpiration rates ($T_{max2}$). In this case, both $T_{max1}$ and $T_{max2}$ are lower than $T_{maxmax}$, therefore, the plateaus of both light blue lines are determined by the available radiation. Energy also influences the response of the plant





**Table 2.** Parameters for the $C_3$ photosynthesis model.

| Parameter | Value | Description |
|---|---|---|
| $H_{Kc}$ [J mol$^{-1}$] | 59430 | Activation energy for $K_c$ |
| $H_{Ko}$ [J mol$^{-1}$] | 36000 | Activation energy for $K_o$ |
| $H_{vV}$ [J mol$^{-1}$] | 116300 | Activation energy for $V_{c,max}$ |
| $H_{dV}$ [J mol$^{-1}$] | 202900 | Deactivation energy for $V_{c,max}$ |
| $H_{vJ}$ [J mol$^{-1}$] | 79500 | Activation energy for $J_{max}$ |
| $H_{dJ}$ [J mol$^{-1}$] | 201000 | Deactivation energy for $J_{max}$ |
| $J_{max_0}$ [$\mu$mol m$^{-2}$ s$^{-1}$] | $2 \times V_{c,max0}$ | Eq. A5 (Kattge and Knorr, 2007) |
| $K_{c_0}$ [$\mu$mol mol$^{-1}$] | 302 | Michaelis constant for $CO_2$ at $\theta_0$ |
| $K_{o_0}$ [$\mu$mol mol$^{-1}$] | 256 | Michaelis constant for $O_2$ at $\theta_0$ |
| $o_i$ [mol mol$^{-1}$] | 0.209 | Oxygen concentration |
| $R_g$ [J mol$^{-1}$ K$^{-1}$] | 8.31 | Universal gas constant |
| $S_v$ [J mol$^{-1}$ K$^{-1}$] | 650 | Entropy term |
| $\theta_0$ [K] | 293.2 | Reference temperature |
| $V_{c,max_0}$ [$\mu$mol m$^{-2}$ s$^{-1}$] | 50 | Eq. A3 |
| $\gamma_0$ [$\mu$mol mol$^{-1}$] | 34.6 | $CO_2$ compensation point at $\theta_0$ |
| $\gamma_1$ [K$^{-1}$] | 0.0451 | Eq. 4 |
| $\gamma_2$ [K$^{-2}$] | 0.000347 | Eq. 4 |





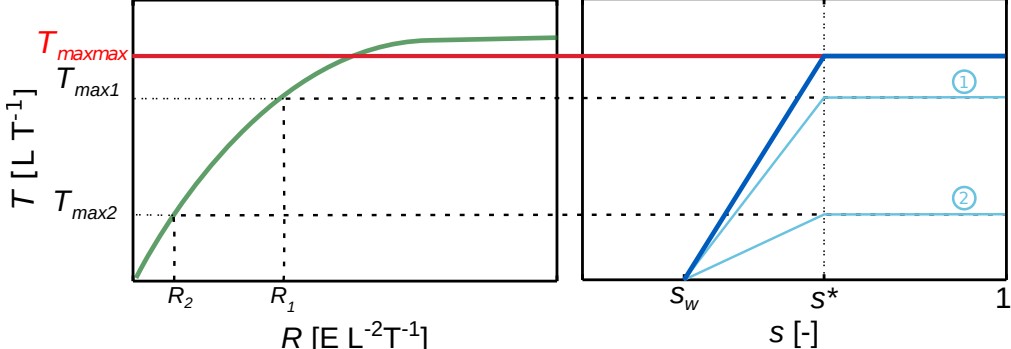

**Figure 1.** Energy (left panel) and water (right panel) dependence of transpiration ($T$). The red line indicates the maximum transpiration rate allowed by the vegetation physiology ($T_{maxmax}$), the green line the relation between available radiation and transpiration rate under no water limitations, the dark blue line the relationship between soil moisture and transpiration rate under no energy limitations, and the light blue lines the relationship of soil moisture ($s$) and transpiration under energy limitations. Light blue line 1(2) shows the behavior of $s$ corresponding to the $T_{max1}$ ($T_{max2}$) resulting from the radiation value $R_1(R_2)$. Soil moisture for values greater than the incipient stomata closure soil moisture ($s^*$) is limited by the maximum transpiration rate ($T_{maxmax}$ or $T_{max}$), while for values between the wilting point ($s_w$) and $s^*$, limitations are given by the water availability.

to water stress (Petersen et al., 1991, 1992). The rate of water loss is proportional to the water vapor concentration gradient
within the vegetation and the bulk atmosphere (Pallardy, 2008), and high radiation values result in high vapor–pressure deficit
in the air. When there is much energy in the atmosphere, vegetation reacts more drastically to water stress ($s < s^*$) because it
can lose water at a high rate (see the steeper light blue line 1 from $s^*$ to $s_w$ in the right panel of Fig. 1). Vegetation begins to
rapidly close their stomata as soil moisture decreases, reducing its transpiration from $T_{max1}$ when $s > s^*$ to zero when $s < s_w$.
On the other hand, when the energy demand is low ($R_2$), vegetation can also suffer water stress, but its reaction may be slighter
(Kaufmann, 1976), as shown in the light blue line 2 with $T_{max}$ equal to $T_{max2}$.

## 3.1 Water–limited ecosystems

The water–limited regime occurs when $ET$ is very sensitive to $s$. This regime is associated with arid and semiarid ecosystems
(Budyko, 1974; Eagleson, 1982; Seneviratne et al., 2010). Water restricts $ET$ by its scarcity, intermittency, and unpredictability
(Porporato and Rodríguez-iturbe, 2002), and photosynthesis is controlled by soil moisture (Porporato and Rodríguez-iturbe,
2002; Daly et al., 2004).

When soil moisture decreases, vegetation reduces its stomata aperture avoiding changes in its internal water status (Cowan
and Farquhar, 1977; Lhomme, 2001). Stomata close as a response to a signal from the roots when the soil is dry before leaf
wilting (Schulze, 1986). This phenomenon is known as vegetation water stress and occurs because vegetation needs an adequate
level of humidity in their tissues to growth and survival (Davies et al., 1990; Lhomme, 2001). The description and effects of
water stress are thoroughly discussed by Schulze (1986); Davies et al. (1990); Flexas and Medrano (2002); Chaves et al. (2003);



Xu et al. (2010); Tardieu et al. (2018); Sloan et al. (2021), among others. Laio et al. (2001) proposed a transpiration model as a function of soil moisture for arid and semiarid regions. In this model, there are no energy limitations, and is expressed as:

$$T(s) = \begin{cases} 0, & 0 < s \leq s_w \\ T_{max}\frac{s-s_w}{s^*-s_w}, & s_w < s \leq s^* \\ T_{max}, & s^* < s \leq 1. \end{cases} \quad (1)$$

The term $T_{max}$ represents the maximum transpiration from the vegetation in the presence of unlimited water and energy.
When $s < s^*$, $T$ is assumed to decrease linearly because of the limitations of soil moisture until it reaches the wilting point, $s_w$. Below $s_w$ transpiration ceases. The right panel of Fig.1 represents the behavior of transpiration as modeled by Eq. 1.

### 3.2 Energy–limited and seasonal ecosystems

The energy–limited regime occurs when soil moisture is most of the time greater than a critical value, with $ET$ weakly dependent on $s$ (Budyko, 1974; Seneviratne et al., 2010). This regime is associated with wet ecosystems. Light limits by its
high spatiotemporal variability, that is related to structural and environmental heterogeneity (gapping and clumping of foliage, gaps in the canopy, leaf orientation, type and distribution of clouds, topography, seasonal trends in plant phenology, and seasonal movements of the sun) (Baldocchi and Collineau, 1994).

Radiation in the spectral band of photosynthetically active radiation (PAR) directly drives the fundamental plant physiological processes involved in transpiration, i.e., photosynthesis, assimilation, and respiration. Besides, it indirectly influences
secondary processes such as plant growth, seedling generation, structure, and gas emission (Monteith, 1965; Baldocchi and Meyers, 1991; De Pury and Farquhar, 1997).

Transpiration and photosynthesis are processes take place simultaneously as vegetation losses water by transpiration while taking up $CO_2$ for photosynthesis (Daly et al., 2004; Yu et al., 2004). Photosynthetic rate is a function of irradiance, $CO_2$ concentration, temperature, nutrients and, water supply (Luoma, 1997). However, under well-watered conditions, PAR is one
of the major environmental factors controlling photosynthesis, stomatal conductance, and consequently, transpiration, in a great number of species (Kaufmann, 1976; Schulze et al., 1995; Mielke et al., 1999; Gao et al., 2002). Stomatal conductance and transpiration increase with PAR (Gao et al., 2002; Pieruschka et al., 2010), as shown in the left graph of Fig. 1. This can be explained by the proportionality between the potassium cation concentration in guard cells and PAR. An increase in the potassium cation concentration causes a decrease in the osmotic potential of guard cells, with additional water moving from
the epidermal cells to the guard cells. This provokes great turgor pressure inside guards and reduces turgor on subsidiary cells so that vegetation opens its stomata, rising thus its conductance and transpiration (Cooke et al., 1976; Gao et al., 2002; Yu et al., 2004). In seasonal ecosystems, the availability of water and energy fluctuates, and vegetation can present unique adaptations and effects on the hydrological cycle that differ between water–limited and energy–limited ecosystems (Asbjornsen et al., 2011).



## 3.3 Transpiration and available energy

Available energy affects transpiration, stomatal aperture and photosynthesis through light receptors driving $CO_2$ fixation and lower intercellular $CO_2$ concentration (Yu et al., 2004), and determines the diabatic component of transpiration (Monteith and Unsworth, 2013). Hence, to properly study the effects of radiation on transpiration (T), the relations among carbon assimilation ($A_n$), stomatal conductance ($g_s$) and transpiration must be taken into account. For this, the Penman–Monteith equation, the Leuning's stomatal conductance model, the Farquhar model, and a simplified energy balance model are solved numerically and simultaneously. The model is run at a half-hourly scale (resolution of the FLUXNET database), but is integrated on a daily scale since we are extending the model of Laio et al. (2001) of the same time resolution. Bartlett et al. (2014), Daly et al. (2004) and Leuning et al. (1995) present methodologies to solve simultaneously stomatal conductance, $CO_2$ assimilation, and the energy balance.

Penman–Monteith equation (Monteith, 1965; Monteith and Unsworth, 2013) is adopted because it is a widely used model that relates transpiration and stomatal conductance. It is expressed as:

$$T = \frac{(\rho_a c_p D g_{ba} + \Delta_e R) \, g_s LAI}{\rho_w \lambda_v \left[ \Delta_e g_s LAI + \gamma_p \left( g_{ba} + g_s LAI \right) \right]}, \tag{2}$$

where $\lambda_v$ is the latent heat of vaporization (2.26 MJ kg$^{-1}$), $\rho_w$ and $\rho_a$ are the water (997 kg m$^{-3}$) and air (1.2 kg m$^{-3}$) densities, respectively, $c_p$ is the specific heat of air (1.013·10$^{-3}$ MJ kg$^{-1}$ K$^{-1}$), $\Delta_e$ is the slope of the saturation of vapor pressure as a function of temperature, $\gamma_p$ is the psychometric constant, $D$ is the saturation vapor pressure deficit, LAI is the leaf area index, and $g_{ba}$ is the series of leaf boundary conductance ($g_b$) and atmospheric boundary layer conductance ($g_a$). Both $g_a$ and $g_b$ are assumed to be constant. We assume that the available energy coincides with the value of PAR. The term $\rho_a c_p D g_{ba}$ in Eq. 2 is the adiabatic component that accounts for the atmospheric saturation deficit, and the term $\Delta_e R$ is the diabatic component of latent heat loss, related to radiation supply. According to the Penman–Monteith equation, $T$ increases linearly with $R$ and the atmospheric saturation deficit. As $g_{ba}$ is strongly related to wind speed, when it increases, $T$ also increases, and when variables in the numerator remain constant, $\Delta_e$ increases with temperature.

The expression of transpiration of Laio et al. (2001) (Eq. 1) manages to describe the daily $T$ dynamics in energy–limited and seasonal ecosystems provided that $T_{max}$ is defined taking into account the available energy, and its stationarity (i.e., the parameters describing the available energy are fixed during the growing season or climate season). Fig. 2 represents transpiration as a function of soil moisture and available energy ($T(s, R)$) for a particular set of parameter values, using the Penman–Monteith equation and varying radiation from 0 to 18 MJ m$^{-2}$ (for a fixed stomatal conductance). This figure shows that when the available radiation is high, the rate at which transpiration decreases with $s$ is much steeper than when radiation is low, representing the response of vegetation to atmospheric demand. Note that for $R = 0$ there is still a minimal evapotranspiration due to the nonzero value of the adiabatic term.





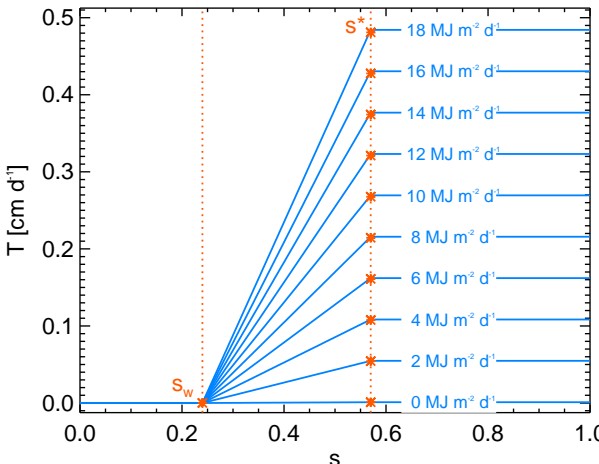

**Figure 2.** Transpiration rate as a function of soil moisture and available radiation according to the Penman–Monteith equation and Laio et al. (2001) model. Each horizontal line represents the available radiation value displayed on it. The parameters used in this figure are $s_w$=0.24, $s^*$=0.57, $\Delta_e$=0.17 kPa $°C^{-1}$, $\gamma_p$=0.06 kPa $°C^{-1}$, LAI=1.4, $g_{ba}$=0.01 m $s^{-1}$, $g_s$=0.004 m $s^{-1}$, and $D$=0.006 kPa.

### 3.3.1 Stomatal conductance

Stomatal conductance ($g_s$) can be calculated using physiological and biochemical models (e.g. Jarvis, 1976; Farquhar et al., 1980; Ball et al., 1987; Farquhar, 1989; Collatz et al., 1991; Leuning, 1995; Gao et al., 2002; Dewar, 2002; Tuzet et al., 2003; Yu et al., 2004). The models most widely used are those based on Jarvis (1976) (e.g. Baldocchi and Meyers, 1991; Peters-Lidard et al., 1997; Daly et al., 2004; Yu et al., 2004) and Ball et al. (1987) (e.g. Leuning, 1990, 1995; Leuning et al., 1995; Daly et al., 2004) approaches.

Net assimilation and transpiration are processes coupled with the stomatal aperture. Therefore, to analyze the dynamics of transpiration, a stomatal conductance model that relates transpiration to net assimilation is required. For this purpose, we use the semi-empirical formulation given by Ball et al. (1987) and improved by Leuning (1990, 1995), expressed as:

$$g_s = 1.6a_1 \frac{A_n}{(c_s - \Gamma^*)\left(1 + \frac{D}{Dx}\right)}. \tag{3}$$

This equation gives $g_s$ in terms of carbon assimilation ($A_n$), water vapor saturation deficit ($D$), $CO_2$ compensation point ($\Gamma^*$), carbon concentration at the leaf surface ($c_s$), a fitted parameter representing the sensitivity of stomata to changes in $D$ ($D_x$), and an empirical constant with a typical value around 15 ($a_1$). The $CO_2$ compensation point is the $CO_2$ concentration at which the $CO_2$ uptake rate in the photosynthesis equals the $CO_2$ loss rate of respiration (Birmingham and Colman, 1979). $\Gamma^*$ is significantly affected by leaf temperature, and according to Brooks and Farquhar (1985), they can be related by:

$$\Gamma^* = \gamma_0 + \left[1 + \gamma_0\left(\theta_l - \theta_0\right) + \gamma_2\left(\theta_l - \theta_0\right)^2\right], \tag{4}$$



where $\gamma_0$, $\gamma_1$ and $\gamma_2$ are empirical constants, $\theta_0$ is the reference temperature, and $\theta_l$ is the leaf temperature.

### 3.3.2 Energy balance

When solving Eqs. 2 and 3 there are three unknowns ($T$, $g_s$ and $\theta_l$), so it is mandatory to involve another equation that allows solving the system. In this case, the energy balance equation:

$$\theta_l = \theta_a + \frac{R - \rho_w \lambda_w T}{c_p \rho_a g_a}. \tag{5}$$

### 3.3.3 Net carbon assimilation

The Farquhar model (Farquhar, 1973; Cowan and Farquhar, 1977; Farquhar et al., 1980) is applied to calculate $A_n$ in sites lacking measurements. This is the most frequently used model to quantify the responses of $C_3$ plants to external perturbations under well-watered conditions. The biochemical demand for $CO_2$ is determined as a function of the photosynthetic photon flux

density (FFPD), $CO_2$ concentration in the mesophyll cytosol ($c_i$) and leaf temperature ($\theta_l$), and expressed as:

$$A_n = f\left(PPFD, c_i, \theta_l\right) = \min\left[A_c, A_q\right], \tag{6}$$

where $A_c$ and $A_q$ are the photosynthesis rates limited by the Ribulose bisphosphate carboxylase–oxygenase (Rubisco) activity, and by the Ribulose bisphosphate (RuP$_2$) regeneration through electron transport, respectively (see Appendix A for more details).

### 3.3.4 Upscaling from half-hourly to daily timescale

The results obtained with the models of transpiration, stomatal conductance, and net assimilation have the temporal resolution of FLUXNET data, i.e, half-hour. To evaluate the daily dynamics of transpiration, we integrate both the calculated results and the information from the FLUXNET database at this time scale. The daily values of $s$, $T$ and $g_s$ correspond to the average during the day, while PAR and $A_n$ are the cumulative sub-daily values. The results of DE–Hai (Germany) and GF–Guy (French

Guiana) are shown as illustrative cases.

## 4 Soil moisture dynamics

Rodríguez-Iturbe et al. (1999) proposed a daily stochastic zero-dimensional model for soil moisture dynamics at a point in terms of climate–soil–vegetation interactions, under seasonally fixed conditions. The stochastic behavior of rainfall propagates through interception, evapotranspiration, runoff, leakage and soil moisture. Rainfall is modeled as a marked Poisson process

that generates infiltration into the soil as a function on the existing soil water content until it reaches saturation ($s = 1$). Soil





water losses are due to evapotranspiration and leakage, which also depend on the soil moisture state. Soil moisture dynamics is the result of the water mass balance over the plant's rooting depth, expressed by the differential equation:

$$nZ_r\frac{ds\left(t\right)}{dt} = \varphi\left[s\left(t\right),t\right] - \chi\left[s\left(t\right),R\left(t\right)\right],\qquad(7)$$

where $n$ is the soil porosity, $Z_r$ is the rooting depth, $s$ is the soil water content, $R$ is the available radiation, $\varphi\left[s\left(t\right),t\right]$ is the infiltration rate, and $\chi\left[s\left(t\right),R\left(t\right)\right]$ is the soil moisture loss rate.

Infiltration is a stochastic component, expressed as:

$$\varphi\left[s\left(t\right),t\right] = P\left(t\right) - I\left(t\right) - Q\left[s\left(t\right),t\right],\qquad(8)$$

where $P\left(t\right)$ is the rainfall rate, $I\left(t\right)$ is the rainfall rate intercepted by the canopy, and $Q\left[s\left(t\right),t\right]$ is the rate of surface runoff generation.

Soil water losses are evaporation, transpiration and leakage, and thus total water loss rate ($\chi$) is given by:

$$\chi\left[s\left(t\right),R\left(t\right)\right] = ET\left[s\left(t\right),R\left(t\right)\right] + L\left[s\left(t\right)\right],\qquad(9)$$

where $ET\left[s\left(t\right),R\left(t\right)\right]$ and $L\left[s\left(t\right)\right]$ are the evapotranspiration and leakage rates, respectively.

$ET$ is modeled as the sum of evaporation ($E$) and transpiration ($T$). $E$ is a fixed rate equal to $E_w$ when $s_w \leq s \leq 1$, which decreases from $s_w$ until it reaches the hygroscopic point ($s_h$), where it becomes zero. Transpiration is modeled as Eq. 1, with
$ET$ given as:

$$ET\left(s\right) = \begin{cases} 0, & 0 < s \leq s_h \\ E_w\frac{s-s_h}{s_w-s_h}, & s_h < s \leq s_w \\ E_w + \left(E_{max} - E_w\right)\frac{s-s_w}{s^*-s_w}, & s_w < s \leq s^* \\ E_{max}, & s^* < s \leq 1. \end{cases}\qquad(10)$$

$E_{max}$ is equal to $T_{max} + E_w$. Appendix B describes the modeling of the other variables in Eqs. 8 and 9.

Following Rodríguez-Iturbe et al. (1999) and Laio et al. (2001), the probability density function (pdf) of soil moisture ($f\left(s\right)$) under steady-state conditions may be derived from the Chapman–Kolmogorov forward equation. The general form of
the solution is:

$$f\left(s\right) = \frac{C}{\rho\left(s,R\right)}\mathrm{e}^{-\gamma s + \lambda' \int \frac{du}{\rho\left(u\right)}}, \text{ for } s \geq s_h,\qquad(11)$$





where $\lambda'$ is the mean time between rainy days, $\gamma = \frac{nZ_r}{\alpha}$ being $\alpha$ the mean rainfall depth of rainy days, $\rho$ is the sum of the soil moisture losses ($ET$ and $L$) normalized by $nZr$, and $C$ is a constant that can be obtained by imposing the condition $\int_{s_h}^{1} \rho(s)\,ds = 1$. This constant is easily obtained numerically, and its analytical expressions are given in Laio et al. (2001) and

Rodríguez-Iturbe and Porporato (2004). Details of the derivation of $f(s)$ can be found in Rodríguez-Iturbe et al. (1999); Laio et al. (2001); and Rodríguez-Iturbe and Porporato (2004). The general solution is:

$$
f(s) = \begin{cases}
\frac{C}{\eta_w}\left(\frac{s-s_h}{s_w-s_h}\right)^{X_1-1} \mathrm{e}^{-\gamma s} & s_h < s \leq s_w \\
\frac{C}{\eta_w}\left[1+\left(\frac{\eta}{\eta_w}-1\right)\frac{s-s_w}{s^*-s_w}\right]^{X_2-1} \mathrm{e}^{-\gamma s} & s_w < s \leq s_{cr} \\
\frac{C}{\eta}\mathrm{e}^{-\gamma s+\frac{\lambda'}{\eta}(s-s^*)}\left(\frac{\eta}{\eta_w}\right)^{X_2} & s_{cr} < s \leq s_{fc} \\
\frac{C}{\eta}\mathrm{e}^{-(\beta+\gamma)s+\beta s_{fc}}\left(\frac{\eta\mathrm{e}^{\beta s}}{(\eta-m)\mathrm{e}^{\beta s_{fc}}+m\mathrm{e}^{\beta s}}\right)^{X_3+1} \\
\quad\cdot\left(\frac{\eta}{\eta_w}\right)^{X_2-1}\mathrm{e}^{X_4} & s_{fc} < s \leq 1,
\end{cases}
\tag{12}
$$

where

$$
X_1 = \lambda'\frac{s_w-s_h}{\eta_w}, \qquad X_2 = \lambda'\frac{s^*-s_w}{\eta-\eta_w}, \qquad X_3 = \frac{\lambda'}{\beta(\eta-m)}, \qquad X_4 = \lambda'\frac{s_{fc}-s^*}{\eta}
$$


$$
\eta_w = \frac{E_w}{nZ_r}, \qquad \eta = \frac{E_{max}}{nZ_r}, \qquad m = \frac{k_s}{nZ_r\left[\mathrm{e}^{\beta(1-s_{fc})}-1\right]}.
$$

$k_s$ is the saturated hydraulic conductivity and $\beta$ is a fitting coefficient that depends on soil type, and $s_{fc}$ is the field capacity. As mentioned before, the transpiration model of Laio et al. (2001) manages to describe the daily $T$ dynamics in energy–limited ecosystems. Consequently, Eq. 10 manages to represent the evapotranspiration dynamics, and Eq. 12 the dynamics of soil

moisture. This is proper as long as $T_{max}$ (and $E_{max}$) is defined as a function of the available energy, and the stationarity of the parameters describing rainfall and radiation is preserved. It is worth noting that considerations in the model of Rodríguez-Iturbe et al. (1999) must continue to be valid, e.g., deep water table, soil homogeneity, distribution of infiltration volume into the rooting depth, etc.

Interactions between vegetation and water table are not considered. This is a realistic assumption for water–controlled arid
and semiarid ecosystems, but may be a questionable one for energy–limited ecosystems. In the latter case, there may exist a close interaction between transpiration and the water table level (Tamea et al., 2009), but this may or may not impact heavily the pdf of soil moisture in systems that are both water– and energy–limited.

## 5 Daily dynamics

Figure 3 shows the relationship between available energy and $CO_2$ assimilation, and available energy and the stomatal con-
ductance in Germany (DE–Hai) and French Guiana (GF–Guy). As DE–Hai (Fig. 3(a,b)) is located in the extratropics, the




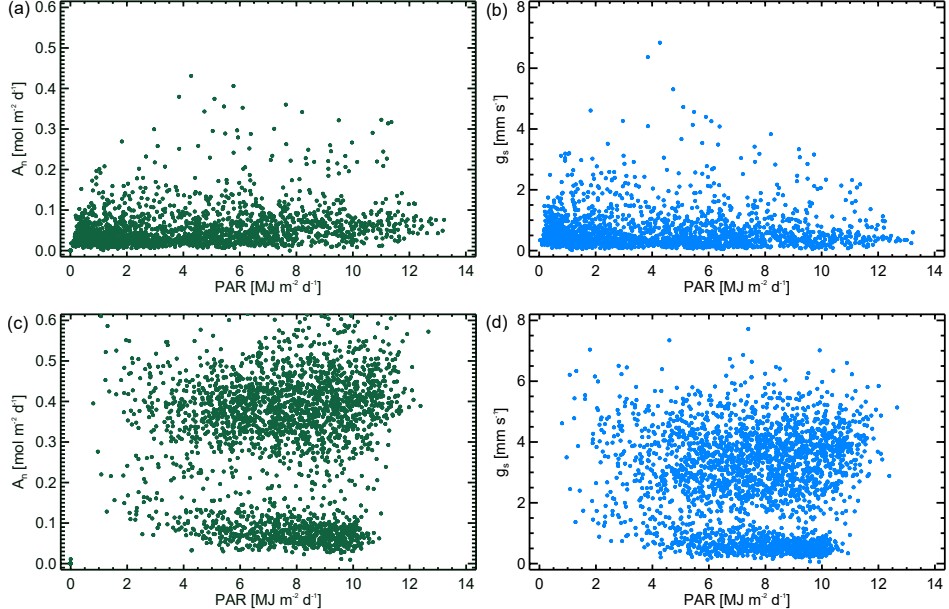

**Figure 3.** Relationship between daily PAR and $CO_2$ assimilation (left panel) and daily PAR and stomatal conductance (right panel) at GE–Hai (a,b) and GF–Guy (c,d).

relationships of PAR and $A_n$, and PAR and $g_s$ are positive for low values of PAR ($\approx$ 4 MJ m$^2$) and negative for high values. The above can be explained by the photo-inhibition phenomenon that occurs under strong light to avoid the destruction of the plant tissues. This phenomenon involves the direct diversion of superfluous radiation energy from the photosystems via fluorescence, and above as heat (Larcher, 1995). Nonetheless, at tropical sites as GF–Guy (see Fig. 3(c,d) ), the relationships

of PAR with $g_s$ and $A_n$ seem more irregular, which can be explained by adaptation strategies developed by vegetation under high radiation throughout the year. Of note is that the PAR values analyzed correspond to those reaching the ground surface, and not those absorbed by vegetation.

Fig. 4 shows the relationship between PAR and transpiration at the same sites shown in Fig. 3. In both types of ecosystems the relationship is direct since when PAR increases, both the adiabatic and diabatic terms of Penman–Monteith increase. Radiation

affects temperature, and this, in turn, modifies the vapor saturation deficit (Zhu et al., 2022). Furthermore, if energy were available, the stomata would open as they could fix more $CO_2$, leading the plant to lose water. However, as shown in Fig. 3, the relation between PAR and $g_s$ is not always direct, since $g_s$ stabilizes at a point (light–saturated plateau) (Lambers et al., 2008), and may even decrease. The effect of light–saturation is also observed on $T$, but not to the point of photo-inhibition, at least for the measured values of PAR used in this analysis.

$A_n$, and consequently $T$, can be limited by many external factors such as PAR, vapor pressure deficit, atmospheric pressure, soil moisture, and air temperature (Sloan et al., 2021; Cong et al., 2022). Under no external limitations, vegetation transpires as much as its physiology allows. The transpiration rates represented by the points shown in Fig. 4 can be limited by any external





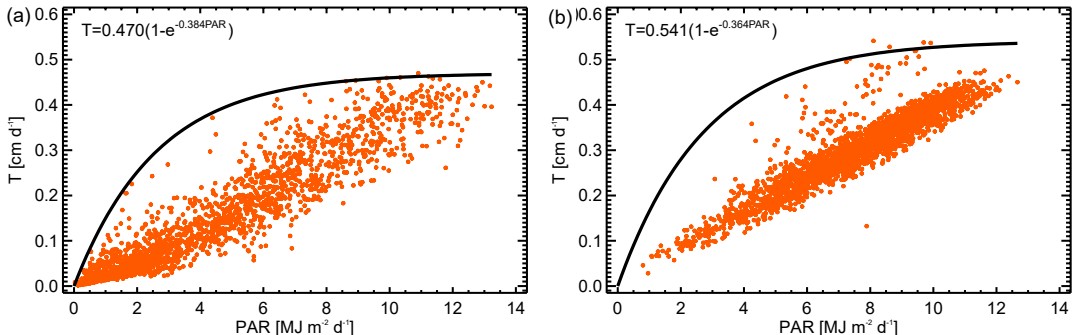

**Figure 4.** Relation of daily PAR and $T$ in DE–Hai (a) and GF–Guy (b). The black line represents the proposed model to relate both variables.

variable, so a link between $T$ and PAR when there are only internal limitations is given by the envelope of these points. The black lines in Fig. 4 indicate these envelopes, which fit well in the studied sites to the expression:

$$T_{max}(PAR) = T^* \left(1 - \mathrm{e}^{-a_T PAR}\right). \tag{13}$$

This expression is a function of the transpiration value at light–saturation ($T^*$) and of a fitting parameter that determines the shape of the curve ($a_T$). This relationship avoids considering the indirect effects of radiation on transpiration ($g_s, \theta_a, D$, etc.). From Eq. 13 and considering the transpiration rate given by the vegetation physiology ($T_{maxmax}$), $T_{max}$ can be defined as:

$$T_{max}(PAR) = \begin{cases} T^* \left(1 - \mathrm{e}^{-a_T PAR}\right), & T_{max}(PAR) < T_{maxmax} \\ T_{maxmax}, & T_{max}(PAR) \geq T_{maxmax}. \end{cases} \tag{14}$$

Note that we considered a constant available energy since its stochasticity at the daily scale does not play a fundamental role in soil moisture dynamics under the assumptions of the Rodríguez-Iturbe et al. (1999) model, as shown by Muñoz (2019).

## 6   Sensitivity analysis

Figure 5 shows the response of soil water dynamics to PAR when other parameters of the Rodríguez-Iturbe et al. (1999) and Laio et al. (2001) models vary according to the dimensionless groups:

$$\pi_1 = \frac{E_{max}}{\alpha\lambda}, \qquad \pi_2 = \frac{nZ_r}{\alpha}, \qquad \pi_3 = \frac{k_s}{\alpha\lambda}, \qquad \pi_4 = \frac{k_s}{E_{max}}, \tag{15}$$

where $\alpha$ is the mean rainfall depth of rainy days. These dimensionless groups simplify the visualization and interpretation of results (Bridgman, 1922; Barenblatt, 1996; Gorokhovski and Hosseinipour, 1997; Butterfield, 1999; Barenblatt and Isaakovich, 2003). The sensitivity of the model output to each parameter is evaluated by moving the input parameter within an appropriate





range and keeping the other parameters fixed. $\pi_1$ and $\pi_2$ groups have been adopted in previous works to analyze the soil
moisture response to rainfall forcing, soil and vegetation changes (e.g. Li, 2014; Feng et al., 2012; Daly and Porporato, 2006;
Porpotato et al., 2004; Rodríguez-Iturbe and Porporato, 2004; Guswa et al., 2002; Milly, 2001; Rodríguez-Iturbe et al., 1999;
Milly, 1993). $\pi_1$ is the *dryness index* of Budyko (1974) and represents the ratio between the maximum evapotranspiration rate
and the long-term mean rainfall rate. $\pi_2$ is called the *storage index* and is the ratio between the amount of water that can be
stored in the soil (until the rooting depth) and the long-term mean rainfall depth (Feng et al., 2012). $\pi_3$ and $\pi_4$ are proposed by
Guswa et al. (2002). $\pi_3$ is the *runoff index* and relates the saturated hydraulic conductivity coefficient and the long-term mean
rainfall rate and, $\pi_4$ is the *infiltration index*, relating the saturated hydraulic conductivity and the maximum evapotranspiration
rate.

For this analysis, we consider a loamy sand soil and a grass cover with the parameters in the caption of Fig. 5, which shows
the results of the four dimensionless groups. Fig. 5(a) shows the pdf of $s$ ($f(s)$) for $\pi_1$ values between 0.1 and 1.4. As the value
of $\pi_1$ increases, $f(s)$ moves to the left. Higher $\pi_1$ results in lower soil moisture values in the long-term, since water losses due
to evapotranspiration are greater than soil water gains due to rainfall. High values of available energy result in lower modes and
greater dispersion than low PAR values. Fig. 5(b) shows $f(s)$ for $\pi_2$ varying between 4 and 20, since natural ecosystems tend
to have root zones deep enough to result in values of $\pi_2$ larger than 1.0 (Milly, 2001). The higher the value of $\pi_2$, the lower the
soil moisture. For large values of $nZ_r$, characteristic of plants with deeper roots such as trees, the amount of rainfall reaching
the soil is distributed into a larger volume (according to the model), resulting in smaller increases in $s$. For lower values of $nZ_r$,
rainfall is uniformly distributed in a smaller volume, increasing soil moisture rapidly. Very high and very low $\pi_2$ values occur
when soil storage capacity is much larger or smaller than the rainfall amount, respectively. High PAR changes the dynamics
of $s$, notably for high values of $\pi$ related to large soil water storage or very small rainfall. Fig. 5(c) shows the results for $\pi_3$
values varying between 50 and 400. As the *runoff index* increases, the water moves rapidly out of the soil, decreasing $s$. As
for $\pi_2$, differences in available energy translate into very different soil moisture dynamics for $\pi_3$, especially for high values,
occurring when the amount of water flowing out the soil is much greater than the rainfall rate. Fig. 5(d) shows $f(s)$ for $\pi_4$
values between 100 and 1000. For low values of $\pi_4$, $s$ remains high because water losses are minor. For high values of $\pi_4$
(greater than 550), the mode of the pdfs stabilizes near the field capacity point, changing only its frequency, and consequently,
the dispersion. When $k_s$ is much larger than $E_{max}$, soil loses water by leakage at a very high rate, being the evapotranspiration
and its variability less relevant. High values of PAR result in curves more pulled to the left than low values of PAR.

If the available energy is high (dotted lines), the curves of $f(s)$ for all $\pi$ groups move more rapidly to the left than for low
values (solid lines), since vegetation transpires at higher rates, maintaining soil moisture lower. The sensitivity of $s$ is more
noticeable for $\pi$ values related to lower soil moisture because the energy demand in the atmosphere changes the rate at which
vegetation decreases its transpiration when it is under water stress. The dimensionless groups that consider $E_{max}$ ($\pi_1$ and
$\pi_4$) show less sensitivity to PAR and the modes always a minor frequency for high available energy. The other dimensionless
groups ($\pi_2$ and $\pi_3$) show a more noticeable variation with PAR, completely changing the dynamics of $s$ for some $\pi$ values
(e.g., $\pi_2$=16 and $\pi_3$=225). Furthermore, the mode has a high (low) frequency for low values of PAR when it is greater (lower)
than $s^*$, decreasing (increasing) the dispersion.





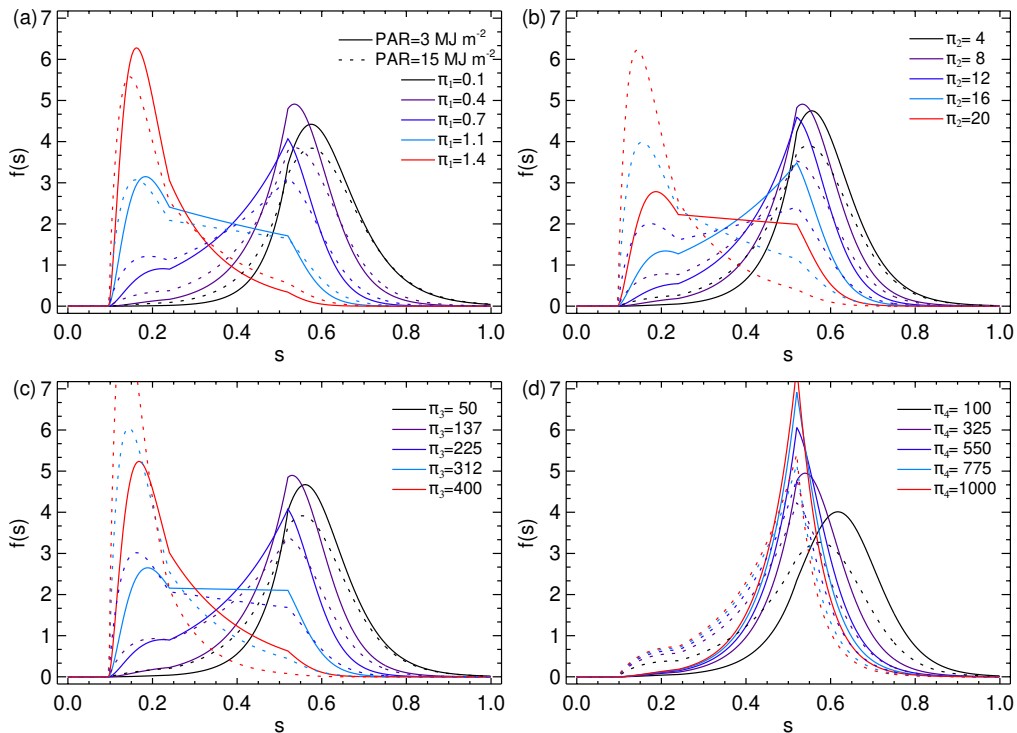

**Figure 5.** Dimensionless sensitivity analysis of soil water dynamics conditioned by available energy using the dryness (a), storage (b), runoff (c), and infiltration (d) indices. Each color corresponds to a value of $\pi$, solid lines represent a low value of PAR (3 MJ m$^{-2}$), and dotted lines a high value (15 MJ m$^{-2}$). Parameters in this figure are $\alpha$=2 cm, $\lambda$=0.5 d$^{-1}$, $\Delta$=0 cm, $Z_r$=30 cm, $T_{max}$=0.47 cm d$^{-1}$, $a$=0.384 m$^2$ MJ $^{-1}$, $b$=4.48, $\beta$=12.7, $n$=0.42, $k_s$=100 cm d$^{-1}$, $s_h$=0.08, $s_w$=0.10, $s^*$=0.24, and $s_{fc}$=0.52.

## 7 Long-term water balance

Figure 6 shows the behavior of the components of the water balance normalized by the average rainfall rate for a loamy sand soil. The expression of each component can be consulted in Laio et al. (2001) and Rodríguez-Iturbe and Porporato (2004). Figs. 6 (a,b) show the influence of rainfall events frequency ($\lambda$) for PAR equal to 3 and 15 MJ m$^2$, respectively. In both cases, the fraction of intercepted water ($I$) is constant and equal, since it changes proportionally to rainfall rate. The percentage of runoff ($Q$) increases with $\lambda$ in a similar proportion for both cases. The fraction of water transpired under stressed

conditions ($E_s$) decreases rapidly until $\lambda \approx 0.3$ d$^{-1}$ for PAR=3 MJ m$^{-2}$ and until $\lambda \approx 0.5$ d$^{-1}$ for PAR=15 MJ m$^{-2}$, being in the first case much lower. The same behavior is observed in the fraction of water transpired under non-stressed conditions ($E_s$). When PAR is low, the percentage of leakage is higher than when PAR is high, and the percentage of evapotranspired water is significantly lower. This suggests that more water reaching the soil is lost by evapotranspiration in water–limited regions than in energy–limited regions (for these parameter values), becoming $Q$ and $L$ more important in energy–limited ecosystems.

These results are in agreement with field observations and results found in previous studies (e.g., Sala et al., 1992; Entekhabi





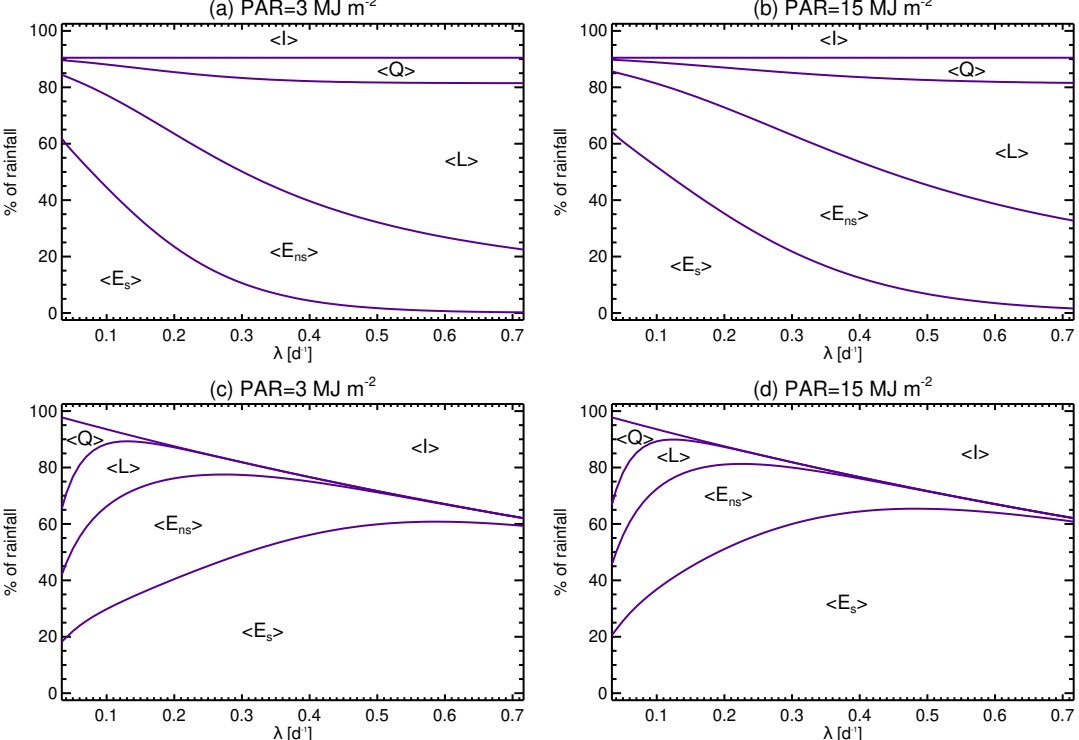

**Figure 6.** Examples of the behavior of the components of the water balance normalized by the average rainfall $\langle P \rangle$ for loamy sand soil, grass vegetation, and PAR=3 MJ m$^{-2}$ (a,c) and PAR=15 MJ m$^{-2}$ (b,d). In (a,b) the parameter $\lambda$ is varied while $\alpha$ is maintained as a constant, and in (c,d) both $\alpha$ and $\lambda$ are varied while maintaining constant the total amount of rainfall during a season. The parameters are shown in caption of Fig. 5

.

and Rodríguez-Iturbe, 1994; Golubev et al., 2001; Rodríguez-Iturbe and Porporato, 2004; Robock and Li, 2006; Roderick et al., 2009).

Figure 6(c,d) shows the behavior of the water balance when $\lambda$ and $\alpha$ are varied while maintaining constant the total amount of precipitation during a season $\Theta$ ($\Theta = \alpha \cdot \lambda \cdot nd$, being $nd$ the number of days of the growing season) for PAR equal to 3 350 and 15 MJ m$^2$, respectively. For this figure $\Theta = 60$ cm and $nd = 200$ d. Interception increases almost linearly with $\lambda$ while runoff decreases rapidly. According to Laio et al. (2001), such decrease depends strongly on the ratio between soil depth and mean depth of rainfall events. The opposite behavior of interception and runoff determines a maximum of evapotranspiration at certain values of $\lambda$. As when only $\lambda$ is varied, the main difference in the behavior of the water balance components for high and low PAR is observed in the percentage of evapotranspiration, being remarkably lower in the first case.



## 8 Conclusions

Rodríguez-Iturbe et al. (1999) and Laio et al. (2001) developed an ecohydrological model to study the probability distribution of soil moisture at a site when both the input into the soil from rainfall and the losses from evapotranspiration and leakage are dependent on the moisture state of the soil. As stated by the authors, the model assumptions make it suitable for water–limited ecosystems. In this paper we have extended the aforementioned models for the case of energy–limited ecosystems. In our extended model, evapotranspiration is a bivariate function of both soil water content and available photosynthetic active radiation (Eqs. 1 and 14).

The analytical expression of our extended model for the probability distribution of soil moisture is the same proposed by Laio et al. (2001) (Eq. 12), as long as $E_{max}$ in $\eta$ is calculated in terms of radiation.

To hold analytical tractability we keep most of the assumptions of the original model, as rainfall stationarity, deep water table, and homogeneous soil and vegetation cover. The extended model assumes also stationarity in the PAR regime.

We also analyzed the daily relationship between transpiration and photosynthetic active radiation by coupling the water and $CO_2$ fluxes through the leaf using data from FLUXNET. As transpiration is directly related to the stomatal conductance, the relation between PAR and $T$ is direct until a certain point where transpiration ceases to increase. We proposed an expression to parameterize the link between these two variables, which allows calculating the daily maximum transpiration rate from the value of daily available energy.

Several examples are presented exhibiting the influence of radiation on $s$ (see Figs. 5 and 6), noticing that the available energy can notoriously change the soil moisture dynamics, and that evapotranspiration plays a more important role in water–limited than in energy–limited ecosystems. We note that these results are only valid on a daily scale since soil–climate–vegetation system dynamics change at finer temporal scales.

## Appendix A: Assimilation model for C$_3$ plants

The photosynthesis rates limited by the Ribulose bisphosphate carboxylase–oxygenase (Rubisco) activity ($A_c$), and by the Ribulose bisphosphate (RuP$_2$) regeneration through electron transport ($A_q$) are given by:

$$A_c = V_{c,max}\left(\theta_l\right) \frac{c_i - \Gamma^*}{c_i + K_c\left(1 + o_i/K_o\right)}, \tag{A1}$$

$$A_q = \frac{J}{4} \frac{c_i - \Gamma^*}{c_i - 2\Gamma^*}, \tag{A2}$$





where $\Gamma^*$ is the $CO_2$ compensation point (see Eq. 4), $o_i$ is the intercellular oxygen concentration, $V_{c,max}$ is the maximum catalytic activity of Rubisco in the presence of saturating levels of $RuP_2$ and $CO_2$ (Eq. A3), and $K_c$ and $K_o$ are Michaelis coefficients for $CO_2$ and $O_2$, respectively, given by Eq. A4.

$$V_{c,max}\left(\theta_l\right)=V_{c,max_0}\frac{\exp\left[\frac{H_{vV}}{R_g\theta_0}\left(1-\frac{\theta_0}{\theta_l}\right)\right]}{1+\exp\left[\frac{S_v\theta_l-H_{dV}}{R_g\theta_l}\right]}, \tag{A3}$$

$$K_x\left(\theta_l\right)=K_{x_0}\exp\left[\frac{H_{Kx}}{R_g\theta_0}\left(1-\frac{\theta_0}{\theta_l}\right)\right]. \tag{A4}$$

$J$ is the electron transport for a given absorbed photon irradiance, and is equal to $\min\left[J_{max}\left(\theta_l\right),PPFD\right]$, being $J_{max}$ equal to:

$$J_{max}\left(\theta_l\right)=J_{max_0}\frac{\exp\left[\frac{H_{vJ}}{R_g\theta_0}\left(1-\frac{\theta_0}{\theta_l}\right)\right]}{1+\exp\left[\frac{S_v\theta_l-H_{dJ}}{R_g\theta_l}\right]}. \tag{A5}$$

The parameters not mentioned here are described in Table 2.

## Appendix B:  Soil moisture model

The variables involved in Eq. 7, except the evapotranspiration (see Eq. 10 in section 4), are modeled as Rodríguez-Iturbe et al. (1999) and Laio et al. (2001).

### B1   Rainfall and interception

Daily precipitation is modeled through a marked Poisson process with arrival rate $\lambda$ (Eagleson, 1972). The pdf of time intervals between rainy days $\tau$ is exponential with mean $\lambda^{-1}$:

$$f_T\left(\tau\right)=\lambda e^{-\lambda\tau},\text{ for }\tau\geq 0. \tag{B1}$$

The marks correspond to the rainfall depth of rainy days, $h$, modeled as an independent exponentially distributed random variable with mean $\alpha$.

$$f_H\left(h\right)=\frac{1}{\alpha}e^{-\frac{1}{\alpha}h},\text{ for }h\geq 0. \tag{B2}$$

The values of $\alpha$ and $\lambda$ are assumed to be time-invariant quantities during the modeling period (growing season or climate
season), i.e. rainfall is considered as a stationary stochastic process.



Rainfall rate is linked to the probability distributions expressed by Eqs. B1 and B2 as the marked Poisson process (Rodríguez-Iturbe and Porporato, 2004):

$$P(t) = \sum_1 h_i \delta(t - t_i),\tag{B3}$$

where $\delta(\cdot)$ is the Dirac delta function, $h_i$ is the sequence of random rainfall depths distributed as Eq. B2 and $[\tau_i = t_i - t_{i-1}, i = 1, 2, 3...]$
is the interarrival time sequence of a stationary Poisson process of frequency $\lambda$.

Following Rodríguez-Iturbe et al. (1999), interception is modeled through a threshold, $\Delta$, such that only rainfall above $\Delta$ reaches the soil. The censored rainfall process is thus Poissonian with rate $\lambda'$:

$$\lambda' = \lambda \int_{\Delta}^{\infty} f_H(h)\,dh = \lambda e^{-\frac{\Delta}{\alpha}}.\tag{B4}$$

The depths $h'$ of the censored rainfall process have the same exponential distribution as the original marks $h$ (Rodríguez-
Iturbe et al., 1999). Then, the new Poisson process is:

$$P(t) - I(t) = \sum_1 h_i' \delta(t - t_i'),\tag{B5}$$

where $[\tau_i' = t_i' - t_{i-1}', i = 1, 2, 3...]$ is the interarrival time sequence of a stationary Poisson process with frequency $\lambda'$.

## B2   Infiltration and runoff

Surface runoff is generated via saturation excess (Dunne mechanism) that occurs when the infiltrated water saturates the soil
profile. When rainfall depth is less than or equal to the available soil water storage, all the water from rainfall infiltrates. Infiltration is thus a function of the amount of rainfall and soil moisture, being a stochastic and state-dependent component. Its magnitude and temporal occurrence are controlled by soil moisture dynamics (Rodríguez-Iturbe and Porporato, 2004). The probability distribution of the infiltration may then be written as (Rodríguez-Iturbe et al., 1999):

$$f_Y(y, s) = \gamma e^{-\gamma y} + \delta(y - 1 - s) \int_{1-s}^{\infty} \gamma e^{-\gamma u}\,du, \text{ for } 0 \leq y \leq 1 - s,\tag{B6}$$

where $\gamma = \frac{nZ_r}{\alpha}$ and $y$ is the dimensionless infiltration normalized by $nZ_r$. Infiltration from rainfall can be written as:

$$\varphi[s(t), t] = nZ_r \sum_1 y_i \delta(t - t_i'),\tag{B7}$$

where $[y_i, i = 1, 2, 3, ...]$ is the sequence of random infiltration events whose distribution is represented by Eq. B6.





### B3 Leakage

Losses by leakage or deep infiltration, $L$, occur when soil water content is higher than field capacity, $s_{fc}$. The maximum

percolation rate equals the saturated hydraulic conductivity, $k_s$, and decreases rapidly when the soil begins to dry, as expressed

by (Laio et al., 2001):

$$L(s) = K(s) = \frac{k_s}{\mathrm{e}^{\beta(1-s_{fc})} - 1} \left[ \mathrm{e}^{\beta(s-s_{fc})} - 1 \right], \text{ for } s_{fc} < s \leq 1. \tag{B8}$$

### B4 Soil-drying process

During no-rain periods, soil moisture decays are deterministically modeled from initial values that depend on the the previous

history of the entire soil–drying–wetting process. The soil moisture losses normalized by $nZ_r$ are:

$$
\begin{aligned}
\rho(s, R_n) &= \frac{\chi(s,R)}{nZ_r} = \frac{E(s,R) + L(s)}{nZ_r} \\
&= \begin{cases}
0, & 0 < s \leq s_h \\
\eta_w \frac{s - s_h}{s_w - s_h}, & s_h < s \leq s_w \\
\eta_w + (\eta - \eta_w) \frac{s - s_w}{s^* - s_w}, & s_w < s \leq s^* \\
\eta, & s^* < s \leq s_{fc} \\
\eta + m \left[ \mathrm{e}^{\beta(s - s_{fc})} - 1 \right], & s_{fc} < s \leq 1.
\end{cases}
\end{aligned} \tag{B9}
$$

### Appendix: List of Symbols

$\rho_a$      Air density [kg m$^{-3}$]

$\alpha$      Mean rainfall depth of rainy days [cm]

$\beta$      Soil fitting coefficient [-]

$\chi$      Soil moisture loss rate [cm d$^{-1}$]

$\Delta$      Interception by canopy threshold [cm]

$\Delta_e$      Slope of the saturation of vapor pressure [kPa K$^{-1}$]

$\eta$      Ratio of maximum evapotranspiration rate and soil available space [d$^{-1}$]

$\eta_w$      Ratio of evaporation rate and soil available space [d$^{-1}$]

$\gamma$      Ratio of available space in the soil and mean rainfall depth [-]





| $\Gamma^*$ | $CO_2$ compensation point [$\mu$mol mol$^{-1}$] |
|---|---|
| $\gamma_0$ | Empirical constant of Brooks and Farquhar model [$\mu$mol mol$^{-1}$] |
| 445 $\gamma_1$ | Empirical constant of Brooks and Farquhar model [K$^{-1}$] |
| $\gamma_2$ | Empirical constant of Brooks and Farquhar model [K$^{-2}$] |
| $\gamma_p$ | Psychometric constant [kPa K$^{-1}$] |
| $\lambda$ | Mean time between rainy days [d$^{-1}$] |
| $\lambda_w$ | Latent heat of vaporization [MJ kg$^{-1}$] |
| 450 $\rho$ | Sum of the soil moisture losses [-] |
| $\rho_w$ | Water density [kg m$^{-3}$] |
| $\tau$ | Time intervals between rainy days [d] |
| $\theta_0$ | Reference temperature [K] |
| $\theta_a$ | Air temperature [K] |
| 455 $\theta_l$ | Leaf temperature [K] |
| $\varphi$ | Infiltration rate [cm d$^{-1}$] |
| $a_1$ | Empirical constant of Leuning's model [-] |
| $A_c$ | Photosynthesis rate limited by Ribulose bisphosphate carboxylase-oxygenase (Rubisco) activity [mol m$^{-2}$ d$^{-1}$] |
| $A_n$ | Net carbon assimilation [mol m$^{-2}$ d$^{-1}$] |
| 460 $A_q$ | Photosynthesis rate limited by Ribulose bisphosphate (RuP$_2$) regeneration through electron transport [mol m$^{-2}$ d$^{-1}$] |
| $a_T$ | Fitting parameter to relate PAR and Transpiration [MJ$^{-1}$ m$^2$ d] |
| $C$ | Constant of soil moisture pdf [-] |
| $c_i$ | $CO_2$ concentration in the mesophyll cytosol [$\mu$mol mol$^{-1}$] |
| $c_p$ | Specific heat of air [MJ kg$^{-1}$ K$^{-1}$] |
| 465 $c_s$ | $CO_2$ concentration at the leaf surface [$\mu$mol mol$^{-1}$] |
| $D$ | Saturation vapor pressure deficit [kPa] |





$D_x$     Empirical constant of Leuning's model [kPa]

$E_{max}$     Maximum evapotranspiration rate for unlimited water [cm d$^{-1}$]

$E_w$     Evaporation rate [cm d$^{-1}$]

$ET$     Evapotranspiration rate [cm d$^{-1}$]

$g_a$     Atmospheric conductance [mm s$^{-1}$]

$g_s$     Stomatal conductance [mm s$^{-1}$]

$g_{ba}$     Series of leaf boundary conductance and atmospheric conductance [mm s$^{-1}$]

$h$     Rainfall depth of rainy days [cm]

$H_{dJ}$     Deactivation energy for $J_{max}$ [J mol$^{-1}$]

$H_{dV}$     Deactivation energy for $V_{c,max}$ [J mol$^{-1}$]

$H_{Kc}$     Activation energy for $K_c$ [J mol$^{-1}$]

$H_{Ko}$     Activation energy for $K_o$ [J mol$^{-1}$]

$H_{vJ}$     Activation energy for $J_{max}$ [J mol$^{-1}$]

$H_{vV}$     Activation energy for $V_{c,max}$ [J mol$^{-1}$]

$I$     Rainfall rate intercepted by canopy [cm d$^{-1}$]

$J$     Electron transport for a given absorbed photon irradiance [mol m$^{-2}$ d$^{-1}$]

$J_{max_0}$     Electron transport capacity at $\theta_0$ [$\mu$mol m$^{-2}$ s$^{-1}$]

$J_{max}$     Electron transport capacity [mol m$^{-2}$ d$^{-1}$]

$K_c$     Michaelis coefficient for $CO_2$ [$\mu$mol mol$^{-1}$]

$K_o$     Michaelis coefficient for $O_2$ [$\mu$mol mol$^{-1}$]

$k_s$     Saturated hydraulic conductivity [cm d$^-$]

$K_{c_0}$     Michaelis constant for $CO_2$ at $\theta_0$ [$\mu$mol mol$^{-1}$]

$K_{o_0}$     Michaelis constant for $O_2$ at $\theta_0$ [$\mu$mol mol$^{-1}$]

$L$     Leakage rate [cm d$^{-1}$]





| | | |
|---|---|---|
| $n$ | Porosity [-] | |
| $P$ | Rainfall rate [cm d$^{-1}$] | |
| $Q$ | Surface runoff rate [cm d$^{-1}$] | |
| $R$ | Solar radiation [MJ m$^{-2}$ d$^{-1}$] | |

$R_g$   Universal gas constant [J mol$^{-1}$ K$^{-1}$]

$s$   Soil moisture [-]

$s^*$   Incipient stomata closure [-]

$S_v$   Entropy term [J mol$^{-1}$ K$^{-1}$]

$s_{fc}$   Field capacity [-]

$s_h$   Hygroscopic point [-]

$s_w$   Wilting point [-]

$T$   Transpiration rate [cm d$^{-1}$]

$t$   Time [d]

$T^*$   Transpiration value at light–saturation [cm d$^-$]

$T_{maxmax}$   Maximum transpiration rate given by the vegetation physiology [cm d$^{-1}$]

$T_{max}$   Maximum transpiration rate for unlimited water [cm d$^{-1}$]

$V_{c,max_0}$   Value of $V_{c,max}$ at $\theta_0$ [$\mu$mol m$^{-2}$ s$^-$1]

$V_{c,max}$   Maximum catalytic activity of Rubisco in the presence of saturating levels of RuP$_2$ and CO$_2$ [mol m$^{-2}$ d$^-$1]

$Z_r$   Rooting depth [cm]

LAI   Leaf area index [-]

PAR   Phosynthetically Active Radiation [MJ m$^{-2}$ d$^{-1}$]

PPFD   Photosynthetic photon flux density [mol m$^{-2}$ d$^{-1}$]

*Author contributions.* EM and AO conceived the idea. EM performed the analyses and drafted the paper. All authors contributed in the methodological design, interpretation of results, and paper preparation.



*Competing interests.*  The authors declare that they have no conflict of interest.

*Acknowledgements.*  This work was supported by funding from the Departamento Administrativo de Ciencia, Tecnología e Investigación de Colombia (Colciencias), and Universidad Nacional de Colombia under programs "Becas de Doctorado Nacionales" and "Convocatoria para el Apoyo al Desarrollo de Tesis de Posgrado de la Universidad Nacional de Colombia 2018", respectively. Additionally, we thank Professor Ignacio Rodríguez–Iturbe for his valuable contributions to the development of this work, and Professor Francesco Laio for his revision and

suggestions.



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
