# Peer review of "Probabilistic soil moisture dynamics of water— and energy—limited ecosystems"

_EGUsphere, 2022_

## Author Comment (AC1)

Dear Reviewer,

We would like to thank you for taking the time to assess our manuscript and for your valuable comments and suggestions. We agree that the manuscript could be restructured to make it easier to follow and that some points should be better explained to avoid confusion. We have analyzed the comments carefully and the detailed responses are presented as follows.

Sincerely,

Estefanía Muñoz
Andrés Ochoa
Germán Poveda

**Major comments**

1. This is a relatively limited extension of an existing model, but interesting enough to warrant a publication in HESS. However, for such a limited innovation, the paper is much too long winding. It can be reduced considerably. Why are so many equations related to the Farquhar model (in the main text and the appendix) provided, while in the end no assimilation is calculated: only soil moisture and water balance components. These could be left out or only the equations presented that are needed to support the arguments.

   Authors' response: Thank you for your comment. We agree with you that the manuscript is too long and we can significantly shorten it. Since Farquhar's model is widely known and published, it is not necessary to rewrite the equations but only cite the original work. Regarding assimilation, to calculate the relationship between the transpiration rate and PAR shown in Fig. 4, we coupled models of assimilation, transpiration and stomatal conductance. We noted that this point is not clear enough in the manuscript, so in the restructuring of the manuscript, we will better organize the methodology, explicitly explaining this point.

2. By the way: do we really need Penman-Monteith? According to Penman Monteith, Figure 2 seems to show that Tmax increases linearly with radiation

and does not saturate? This seems a contradiction. With the exponential function chosen.

Authors' response: You make a valid point that the paper should focus more on explaining the transpiration mechanisms considered. When it is assumed that the other variables considered in the Penman-Monteith equation do not depend on radiation, this equation indicates a linear increase in transpiration rate with radiation (Eq. 2). Penman-Monteith assumes that plants are not damaged by excess energy and that there is no light saturation point, so photosynthesis, and thus, transpiration, will increase when radiation does. To avoid this, we calculated the relationship between transpiration and PAR by coupling the assimilation, transpiration and stomatal conductance models and measurements from the FLUXNET database. This calculation allows us to consider how radiation affects transpiration when multiple factors are involved, such as the physiological capacity of plants to transpire.

3. While the paper is too long, it should also be heavily restructured, A much simpler setup would be the following:
   - Introduction
   - Short recap of the Laio et al model (only Eqs 7, 8, 9, 10, 12)
   - Short review of transpiration under both water and energy limited conditions.
     - Describe Figure 1. Also describe why the T-R or T-PAR relationship is a saturating curve? Is this based on Leunings stomatal conductance model and C3 Farquhar assimilation and Penman monteith? Please explain.
     - Support the chosen form of Tmax(PAR) with flux data (Figure 4). Here the fluxnet dataset can be introduced.
     - Leading to the adaptation of the Lai et al model replacing Tmax with Tmax(PAR)
   - Sensitivity study (Figures 5,6)
   - Validation: (see remark hereafter).
   - Appendices A and B can be removed.

Authors' response: Thank you for your specific and detailed suggestion to make our manuscript simpler and easier to understand. We believe that your proposed structure will make the paper easier to follow and the message

clearer. We will update the structure of the manuscript and explain in detail the points you noted here and in the previous comments.

4. To show the importance of the addition an additional validation step is needed. Since you are looking at fluxnet data, at least qualitatively you should be able to show that the pdfs of soil moisture (or at least evapotranspiration) obtained from your adaption are closer to the observed values at the flux sites than the original ones obtained from Laio et al (all other parameters being equal). I realize that the assumption of stationarity does not hold for the German site due to seasonality, but you could focus on one summer month (July) and one early spring month (April) separately to have a water limited and an energy limited example.

Authors' response: We agree that this could add significant value to our proposed extension. We will assess the availability and quality of soil moisture data at the sites we analyzed from the FLUXNET database and compute histograms for comparison with the pdfs calculated using Laio's model and the extension proposed here.

**Minor comments**

1. Abstract, line 8: sensibility -> sensitivity.

   Authors' response: We will fix the error.

2. Line 25: replace "there are seasonal environments .. fluctuates" with "There are areas where both regimes occur depending on the season.

   Authors' response: We will replace the phrase as you suggested as it better explains the idea.

3. Lines 28-32: I do not understand this part. Why are is situ and remote sensing data and numerical simulations presented as three categories. The type of data used and the way equations are solved are two separate issues.

   Authors' response: Thank you for letting us notice it. We will split the comparison according to the type of data and the ways to solve the equations.

4. Line 33: "from such complex processes". What complex processes are meant here?

   Authors' response: We meant the complex processes involved in the soil moisture dynamics, such as water and energy fluxes among the

atmosphere-soil-vegetation system, anthropic effects, etc. Thanks for pointing out that the current phrase in the manuscript is incomplete and unclear. We will fix it.

5. Line 62, start with: "The remaining part of this paper is organized as follows:"

   Authors' response: We will modify the paragraph that describes the structure of the article starting with this sentence you suggest.

6. Line 92: tappers -> tapers

   Authors' response: We will correct the typo.

7. Lines 260-262: groundwater can have a major impact on the pdf of soil moisture and evaporation. See e.g.:
   https://agupubs.onlinelibrary.wiley.com/doi/full/10.1029/2005WR004696

   https://www.sciencedirect.com/science/article/pii/S0304380010001079

   Authors' response: We will complete the implications of this simplification taking into consideration the papers you mentioned and others.

---

## Author Comment (AC2)

Dear Reviewer,

We would like to thank you for taking the time to review our manuscript and for your valuable comments and suggestions. After reading your comments and those of the other reviewers, we decided to restructure the manuscript as described in the response to comment 4. This is to make it clearer and more concise. Please see below for responses to your comments point by point.

Sincerely,

Estefanía Muñoz
Andrés Ochoa
Germán Poveda

**Major comments**

1. As recognized in some lines of the manuscript (L173) but not elsewhere (e.g., L122), Laio's model does take into account the energy constraint, precisely via ET_{max}. With respect to other sensitivity analyses on ET_{max}, here a standard process-based model or an empirical relationship are used to explain variations in such parameter. But, in the end, in its current form, the manuscript appears a sensitivity analysis. Beyond the somewhat misleading framing of this work, all in all changes to the soil moisture balance due to different radiation levels are modest – something that was already concluded in Daly and Porporato (2006, Water Resources Research). Furthermore, neither Laio's model nor the proposed extension take into account the effect of extremely high soil moisture values.

   Authors' response: Thank you for your comment. Although Rodríguez-Iturbe et al. 1999, 2004 and Laio et al. 2002 mentioned that Emax can be obtained from equations like Penman-Monteith, they do not explicitly state that Emax is constrained by energy availability. According to the Penman-Monteith equation, transpiration, in addition to being a function of radiation, is a function of soil heat flux, air vapour pressure deficit, the specific heat of the air, and air temperature, among other variables. In this work, we analyze and separate the effect of the available energy when these other variables are present, and we

obtained an empirical relationship to relate Emax with PAR. Besides, in a previous work (Muñoz 2019, https://repositorio.unal.edu.co/handle/unal/76885), results indicated that the stochastic behaviour of the radiation does not play an important role in the pdf of s if the Rodríguez-Iturbe model is used, but its mean values do. After this conclusion, and maintaining the other assumptions of Rodríguez-Iturbe et al. and Laio et al., this model is suitable in the energy-limited ecosystem for a fixed climate season as long as the value of the maximum transpiration is given by the radiation amount available at the site. Furthermore, based on the results shown in Figures 5 and 6, under some combination of parameter values, there are substantial differences in soil moisture dynamics when contrasting values of PAR are considered. We will better point out the highlights of the work in the manuscript.

2. More importantly, Laio's model is a stochastic soil moisture model, taking into account the randomness in precipitation timing and amount. Radiation fluctuates too, as also apparent from the data used for the empirical relationship. This work appears not to consider this aspect in any way, despite considering daily data (from the sub-hourly upscaling). I find this incorrect. Moreover, a solution to this problem is available in Daly and Porporato (2006, Water Resources Research). If, instead, the point is to consider the average seasonal radiation, then this should be made clear when applying Penman-Monteith and would mean dropping the empirical radiation-transpiration relationship.

Authors' response: As we mentioned in the response to comment 2., the results of our previous work showed that the stochastic component of radiation (after removing seasonality) does not have a determining effect on the pdf of soil moisture when using the Rodríguez-Iturbe model and under its assumptions. As for seasonality, we did not consider it, but this could be the next step. Although we are assuming a deterministic value for energy availability, the Penman-Monteith equation assumes that transpiration increases linearly with radiation, which might be unrealistic given plant damage by energy excess, $CO_2$ limitations, and physiological constraints (e.g. photoinhibition), etc. Given this, we proposed an empirical equation to relate transpiration and PAR that shows a saturation value in which transpiration does not increase even though energy

availability continues to do so. We will explain this point better in the manuscript to avoid confusion.

3. The relative role of Penman Monteith and the empirical radiation-transpiration relationship remains unclear. From my reading, it seems that one of them would suffice in reaching the goal of linking ET_{max} to radiation (but see point 2 regarding a potential crucial difference).

   Authors' response: Thank you for letting us note that this point is not clear in the manuscript. The Penman-Monteith equation was used to couple transpiration, stomatal conductivity and carbon assimilation using the data from the FLUXNET database. Thus, an empirical relationship was found between transpiration and available energy when multiple factors intervene (e.g. the physiological capacity of plants to transpire, the availability of $CO_2$, etc). Note that radiation not only influences the dynamics of transpiration (Penman-Monteith equation) but also that of assimilation (Farquhar model). We will clarify this in the manuscript.

4. I also found the manuscript difficult to follow. Aside from the role of Penman-Monteith vs the empirical relation (see point 3 above), the structure of the text (and subdivision in sections and subsections) is not intuitive and there are many details reported that appear of low relevance to the questions at hand, or so well established not to require anything beyond a reference (e.g., Table 1 and 2; Appendix A and B). I also note that a large number of references are reported in support of rather general points (e.g., L120), where one or two well-chosen references would suffice and serve the reader better.

   Authors' response: Thanks for pointing this out. We agree that the manuscript needs to be restructured to make it clearer and easier to follow. We will remove Appendices A and B and the citations that are not required. Besides, following the suggestion of another reviewer, we will modify the structure as follows: i) Introduction, ii) short recapitulation of the Laio model, iii) review of transpiration mechanism under water and energy-limited conditions, iv) sensitivity analysis, v) validation, and vi) conclusions. Chapter iii) will include the empirical function to relate transpiration and PAR and the replacement of Emax with Tmax(PAR)+Ew.

**Minor comments**

1. L88: the definition of s

Authors' response: You are right. This definition will be replaced by "s is quantified as the ratio of the volumetric water content (ratio of water volume to soil volume) to soil porosity."

2. L90: the fact that transpiration depends only on maximum stomatal conductance, where (as also apparent from Penman Monteith formula) transpiring biomass and leaf-atmosphere coupling play a role too

Authors' response: Thank you for letting us note that this sentence is incomplete. We will complete it.

- L173 (see point 1 above)

Authors' response: We will give the definition of $E_{max}$ by Rodríguez-Iturbe et al. and Laio et al.

- L239: the pdf of s is obtained under *stochastic* steady state, not steady state. This is an important difference

Authors' response: You are right. We will fix this statement.

- There are also few typos, e.g., lines 92, 137, 730 (and elsewhere).

Authors' response: We will check the spelling, grammar, and punctuation of the entire manuscript.